# Modelling Identity Disturbance: A Network Analysis of the Personality Structure Questionnaire (PSQ)

**DOI:** 10.3390/ijerph192113793

**Published:** 2022-10-24

**Authors:** Georgia Mangion, Melanie Simmonds-Buckley, Stephen Kellett, Peter Taylor, Amy Degnan, Charlotte Humphrey, Kate Freshwater, Marisa Poggioli, Cristina Fiorani

**Affiliations:** 1Clinical and Applied Psychology Unit, University of Sheffield, Sheffield S10 2TN, UK; 2Rotherham Doncaster and South Humber NHS Foundation Trust, Rotherham S61 1HE, UK; 3Clinical Psychology Department, University of Manchester, Manchester M13 9PL, UK; 4Tees Esk and Wear Valleys NHS Foundation Trust, Darlington DL2 2TS, UK; 5Private Practice, 20923 Piacenza, Italy

**Keywords:** identity disturbance, network analysis, personality structure questionnaire

## Abstract

Due to the relevance of identity disturbance to personality disorder this study sought to complete a network analysis of a well validated measure of identity disturbance; the personality structure questionnaire (PSQ). A multi-site and cross-national methodology created an overall sample of *N* = 1549. The global network structure of the PSQ was analysed and jointly estimated networks were compared across four subsamples (UK versus Italy, adults versus adolescents, clinical versus community and complex versus common presenting problems). Stability analyses assessed the robustness of identified networks. Results indicated that PSQ3 (unstable sense of self) and PSQ5 (mood variability) were the most central items in the global network structure. Network structures significantly differed between the UK and Italy. Centrality of items was largely consistent across subsamples. This study provides evidence of the potential network structure of identity disturbance and so guides clinicians in targeting interventions facilitating personality integration.

## 1. Introduction

Whilst identity appears a human universal characteristic, there is no single and universal definition of identity. Terms such as identity, ego, the self and personality have been used interchangeably, with differing interpretations but also with closely connected meanings [1]. Identity is in essence the way an individual sees and summarises themselves and therefore is different to personality traits, which are independent of self-perception [2]. Consolidation of identity has been suggested as a core developmental task of adolescence and results in individuals eventually experiencing themselves as more consistent over time/contexts, displaying stable attitudes/values and being able to generate and then achieve long-term goals [1]. 

Should this identity formation process become disrupted (i.e., especially through the impact of trauma), then this damages the formation of a coherent self [3]. This then limits the ongoing process of identity formation, which is subsequently expressed as identity disturbance in adulthood [4]. Identity disturbance in clinical contexts is believed to be maintained by rapid shifts between highly differentiated states of mind compounding the ongoing sense of fragmentation [5]. Identity disturbance has been evidenced to be present across all personality disorders and is part of the diagnostic criteria for borderline personality disorder [6]. There have been found to be four interacting kinds of identity disturbance in borderline personality disorder; role absorption, painful incoherence, inconsistency of thought/feeling/actions, and lack of commitment [7]. 

A methodological approach with potential to better understand of identity disturbance is network analysis, and this is because of its ability to model identity disturbance as a constellation of interacting ‘symptoms’ that can become self-maintaining [8]. In networks, symptoms are represented as ‘nodes’ and their connections (i.e., ‘edges’) represent pairwise relationships. Centrality quantifies how closely the nodes are interconnected and high centrality nodes exert the greatest influence on the network [8]. Network analyses of clinical measures holds promise as this can identify the nodes most central to the presenting clinical problem and also provide potential areas for intervention. Only two studies have previously explored identity disturbance networks. Identity disturbance was shown to play a role in the structure of BPD-like psychopathology in both clinical and non-clinical samples [9], but a later network analysis found identity disturbance to be central in only the less severe BPD group [10]. 

The 8-item *Personality Structure Questionnaire* (PSQ) was designed to measure identity disturbance and is theoretically grounded in the multiple-self states model (MSSM) of cognitive analytic therapy (CAT) [5]. Three validation studies of the PSQ have been conducted [4,11,12] and collectively show that the PSQ is a single factor scale that has good internal and test–retest reliabilities, has a cut-off of 28 for identifying personality disorder, has construct and discriminant validity and is sensitive to measuring psychotherapeutic change. The current study sought to apply the network approach to the PSQ to explore the structure of identity disturbance in more detail and to also meet the call for a network analysis of a specific identity disturbance measure [10]. A multi-sample network analysis approach was used to improve the stability and generalisability of results. To summarise, the present study aimed to: (a) estimate the overall network structure of the PSQ, (b) compare whether network structures and centrality indices differed according to country, age and presenting problem and finally to (c) assess the accuracy and stability of these networks using bootstrapping methods.

## 2. Method

### 2.1. Ethics, Sample and Power

The study was conducted according to the guidelines of the Declaration of Helsinki and ethical approval was granted (Sheffield UREC reference number: 034903; approval date 1 June 2020). The overall sample (*N* = 1553) consisted of secondary data from six differing sites/studies across the United Kingdom (4 sites) and Italy (2 sites). Two of the UK sites also assessed participants from other countries and continents in their methods (see demographics table) because of their online format. Four of the six studies/samples also contained a control group. Informed consent was obtained from all subjects involved in the studies or sites. Table 1 contains a summary of the methodological features of each of the four United Kingdom and two Italian samples. Participants were aged between 12–70 with a mean age of 28.52 (*SD* = 12.81), calculated from a sample of *n* = 1343 where exact ages were available. An adequate network sample size is dependent on the number of parameters needed to be estimated in the network based on the number of nodes [k]; parameters = k × (k − 1/2 × k). The number of parameters estimated based on the eight-item PSQ therefore would be 36 (8 × 7/2 + 8). Guidelines of participant:parameter ratios suggest a 3:1 ratio [13] and this criterion was met by the subsamples within this study. 

### 2.2. PSQ Measure and Missing Data

The PSQ is an eight-item self-report measure scored on a 1–5 Likert scale [4] that is anchored at each extreme with *very true* (i.e., very true, true, may or may not be true, true and very true). This is because the items are presented as dilemmas in order to enable the scaling to make sense to patients or participants. A key component of CAT theory are ‘dilemma patterns’ whereby patients tend to only exist at relational/behavioural/emotional extremes and then alternate between these extremes when under stress [5]. Individual PSQ items are as follows: PSQ1: *My sense of self is always the same* versus *how I act or feel is constantly changing*. PSQ2: *The various people in my life see me in much the same way* versus *the various people in my life have different views of me, as if I were not the same person.* PSQ3: *I have a stable and unchanging sense of myself* versus *I am so different at different times that I wonder who I really am.* PSQ4: *I have no sense of opposed sides to my nature* versus *I feel I am split between two (or more) ways of being, sharply differentiated from each other.* PSQ5: *My mood and sense of self seldom change suddenly* versus *my mood can change abruptly in ways which make me feel unreal or out of control.* PSQ6: *My mood changes are always understandable* versus *I am often confused by my mood changes which seem either unprovoked or quite out of scale with what provoked them.* PSQ7: *I never lose control* versus *I get into states in which I lose control and do harm to myself and/or others.* PSQ8: *I never regret what I have said or done* versus *I get into states in which I do and say things that I later deeply regret.* Total PSQ scores range between 8–40 and higher scores indicate greater levels of identity disturbance. The data consisted of participants’ item-level responses and full PSQ scores for each participant were calculated. Thus, only participants with complete data (*n* = 1549) were included in the network analysis (i.e., 4 participants in the overall dataset had missing values for one or more PSQ items and were therefore excluded). The PSQ in its Italian version was translated from English into Italian by a professional translator for both the adult and the adolescent samples. The PSQ in the adult sample was checked for face validity by *N* = 4 Psychiatrists and Clinical Psychologists and was used in clinical practice for 1-year in order assess ease of comprehension and use by patients, before being assessed as fit-for-purpose as a research tool [12].

### 2.3. Data Analysis

A network was estimated utilising the full sample (*n* = 1549) and jointly estimated networks made comparisons across subsamples. These comparisons were made between the UK (*n* = 625) vs. Italy (*n* = 521), adults (*n* = 521) vs. adolescents (*n* = 254), clinical (*n* = 769) vs. community (*n* = 780) and complex mental health problems (*n* = 477) vs. common mental health problems (*n* = 1073). Independent samples *t*-tests and Cohen’s *d* effect sizes were used to compare mean PSQ item and total scores between the subsamples, with d scores of 0.2, 0.5 and 0.8 indicating small, moderate and large effect sizes respectively. Participants were defined as having a ‘complex mental health problem’ when the presenting problem recorded in the original datasets were labelled as self-harm, personality disorder, psychosis or an eating disorder. ‘Common mental health problems’ were those originally labelled as presenting with anxiety, depression, trauma and obsessive-compulsive disorder. The nature of the data was such that all adolescent participants were Italian. The age comparison therefore utilised Italian data only, whilst the nationality comparison excluded adolescents, in order to prevent age and nationality acting as confounding variables within network comparisons. 

### 2.4. Network Estimation 

The overall PSQ network in the pooled sample was estimated using a Gaussian Graphical Model, a network in which edges represent partial correlations of ordinal or continuous data (using R packages *qgraph* and *glasso*). Using partial correlations ensures that relationships between nodes are not confounded by relationships with other network variables to enable unbiased centrality analyses. GGMs are usually estimated using the graphical lasso, a method which utilises regularisation to avoid estimation of spurious edges. As a result, sparser and more interpretable networks are obtained in which covariance among nodes is explained with as few edges as necessary. If two nodes are connected by an edge in the resulting graph, then they are statistically related after controlling for all other variables in the network. If no edges are present, then they are conditionally independent. 

The Fused Graphical Lasso, an extension of GGM, was used to estimate networks across the subsamples using the R package *EstimateGroupNetwork*. FGL allows the examination of similarities and differences across different samples and is utilised to produce a more accurate estimation of network structures than estimating networks individually. FGL applies two penalty terms controlled by tuning parameters; firstly, a density penalty and secondly, a penalty on differences among corresponding edge weights in networks computed in different samples. K-fold cross-validation was utilised to select the tuning parameters for the penalty terms; this procedure means that the FGL neither masks differences nor inflates similarities between subsamples. 

In all networks, Polychoric correlations were utilised to calculate edges. Polychoric correlations estimate associations between two variables that are theorised to be continuous and normally distributed but measured on ordinal scales. Following recommendations to ensure the appropriateness of Polychoric correlations, both Polychoric and Spearman’s correlations were initially tested and compared in four steps: plotting the networks and visually inspecting, comparing the minimum and maximum edge-weights in each network, calculating the mean edge-weight in each network and correlating the Polychoric and Spearman edge-weights. These checks indicated that the Polychoric and Spearman’s correlations were similar and thus it was concluded that Polychoric correlations were appropriate.

### 2.5. Network Comparisons 

Networks were estimated and compared between UK versus Italy (adult participants only), adult versus adolescent (Italian participants only), clinical versus community, and complex versus common mental health problems. Participants were represented in multiple networks (i.e., a single participant may be represented in Italian, adult, clinical and complex networks). Comparisons of networks where there are very different sample sizes are difficult to interpret as the level of regularisation is influenced by the sample size. With a smaller sample size, fewer edges are retained. One solution to enable more meaningful comparisons is to compare networks using a data-driven permutation test. The R package *NetworkComparisonTest* was used to statistically compare each pair of networks using this method, and to explore whether all edges were identical between them. Where there were significant differences between networks, post hoc tests were utilised to investigate how many edges were significantly different. *NetworkComparionTest* was also utilised to explore whether global strength estimates, the sum of all absolute edge values, differed between networks. To assess whether differences in sample size were influencing results, due to low power, a sensitivity analysis was conducted applying the Network Comparison Test (NCT) to subsamples with equal sizes. This comparison was achieved by repeatedly subsampling the larger dataset to match the smaller one and repeating the NCT. If this produced different results to the uneven sample size comparisons, then sample size adjusted findings were reported. 

### 2.6. Network Inference and Stability

Centrality metrics were completed to explore which items were most integral to networks. High centrality nodes have strong connections to many other nodes, whilst low centrality nodes are peripheral with fewer and weaker connections. Understanding the connectedness of nodes reflects the clinical relevance of a node. Analysis therefore focused on node strength and predictability as the centrality indices. Betweenness and closeness, two other commonly reported centrality indices are often not reliably estimated, therefore were not included. Node strength is the sum of each edge linked to the node and provides a relative measure of centrality. A node’s predictability is an absolute measure of connectivity and represents the shared variance of each node with its neighbours. Node predictability was calculated using the R package *mgm*. Multiple tests of stability were completed using case dropping bootstrapping methods in the R package *Bootnet*, including bootstrapped edge weights, centrality stability (CS) and edge weight/centrality difference tests. Stability results for the global network of the entire sample are reported in the online Appendix A. The CS coefficient was utilised to assess the stability of centrality indices when observing only portions of the data and is reported for each estimated network (global and all subsamples) in the main results to give an indication of network centrality stability (stability was assessed within each individual subsample network rather than the jointly estimated networks). Data sharing: all the data and the R code for the network analysis are available from the corresponding author on request. 

## 3. Results

### 3.1. Demographics and Sample Description

Demographic characteristics of the full sample are summarised in Table 2. The means for total and item-level PSQ scores by subsample are displayed in Table 3. The UK samples had significantly higher PSQ scores than Italian samples (*t*(1144) = 3.74, *p* < 0.05, *d* = 0.22), adolescents scored significantly higher than adults (*t*(773) = 6.99, *p* < 0.05, *d* = 0.54), clinical samples scored significantly higher than community subsamples (*t*(1547) = 12.21, *p* < 0.05, *d* = 0.62) and participants with complex mental health problems had significantly higher full-PSQ scores compared to common mental health problems (*t*(1548) = −7.50, *p* < 0.05, *d* = 0.41).

### 3.2. Global Network Estimation

Figure 1a depicts the overall estimated network for the full sample. Each node represents a PSQ item whilst edges represent relationships between items, controlling for all other variables. All edges in the global network were positive and thicker edges show stronger associations between nodes. The strongest edges were between PSQ1 and PSQ3 (*changing sense of self* and *unstable sense of self*; 0.34) and between PSQ5 and PSQ6 (*changing moods* and *understandable mood change*; 0.33). Moderate edges were shown between PSQ7 and PSQ8 (*loss of control* and *regret*; 0.22); PSQ3 and PSQ2 (*unstable sense of self* and *others’ views*; 0.21); and PSQ3 and PSQ4 (*unstable sense of self* and *opposed sides of nature*; 0.21). Figure 1b presents the centrality metrics for the overall PSQ network. The most central items in terms of node strength and the sum of edges connected to a node were PSQ3 (*unstable sense of self*) and PSQ5 (*changing moods*). The least central items were PSQ8, (*regret*) and PSQ2 (*others’ views*). Likewise, predictability, the variance of a node explained by its neighbours, was highest for PSQ3 and PSQ5 and lowest for PSQ2 and PSQ8. In sum, PSQ3 (*unstable sense of self*) shared the strongest edge and had the highest node strength and predictability (i.e., indicating that it is the symptom most related to other symptoms in identity disturbance). 

### 3.3. Network Comparison across Subsamples

Figure 2a depicts UK and Italian networks. Both networks display a strong edge between PSQ5 (*changing moods*) and PSQ6 (*understandable mood change*). There was a strong edge between PSQ1 (*changing sense of self*) and PSQ3 (*unstable sense of self*) in the UK network that was not observed in the Italian network. Additionally, there was a moderate edge between PSQ7 (*loss of control*) and PSQ8 (*regret*) in the UK network, which was weaker in the Italian network. As displayed in Figure 2b, overall centrality was quite similar within the country network comparison, with small differences evident in centrality order. PSQ3 (*unstable sense of self*) was the most central node in both the UK (1.65) and Italian (1.97) subsamples in terms of node strength. PSQ2 (*other’s views*) was the least central node in both the UK (−1.39) and Italian (−1.26) subsamples. 

Centrality rank order of five out of the six other PSQ items differed between the UK and Italian samples. In particular, PSQ1 (*changing sense of self*) was less central in the Italian network (ranked 5th) than in the UK network (ranked 3rd), whereas PSQ6 (*understanding mood change*) was more central in the Italian network (ranked 3rd) than in the UK network (ranked 6th). The UK subsample had the highest mean predictability of all subsamples at 0.46 (i.e., average amount of variance of a node is explained by its neighbours in the network was 46%). Predictability in the Italian subsample was 0.37. PSQ3 had the highest predictability in both networks, however rank order of node predictability differed between subsamples for all other nodes. The NCT showed that the edge weights in the UK and Italian subsamples were significantly different (*p* < 0.05). Post hoc tests identified that 3 of 36 pairs of edges (8%) significantly differed and these were PSQ1-PSQ3, PSQ6-PSQ8 and PSQ7-PSQ8. Global strength estimates did not significantly differ between the UK and Italy (UK = 3.37, Italy = 3.24, *p* = 0.15). Further comparison, adjusted for equal sample size, produced consistent results.

The supplementary online results depict the remaining subsample comparisons. In terms of the networks for adults and adolescents, there were differences in centrality order between subsamples. In the adult subsample, PSQ3 (*unstable sense of self*; 2.02) was most central, whereas PSQ5 (*changing moods*; 1.38) was most central in the adolescent subsample. The adolescent subsample had the lowest predictability (0.28) of all of the subsamples. Predictability in the adult subsample was 0.37. The NCT showed that there was no statistically significant difference between the adult and adolescent networks (*p* = 0.33). The clinical and community networks were highly similar. PSQ3 (*unstable sense of self*) was the most central node for both clinical and community participants (1.52; 1.88 respectively), whilst PSQ2 (*others’ views*) was the least central in clinical and community subsamples (−1.26; −1.19). Predictability was similar between subsamples at 0.40 in the clinical subsample and 0.32 in the community subsample. The NCT showed that the community and clinical subsamples were not significantly different from each other (*p* = 0.80). Further analysis, adjusted for sample size, produced results consistent with the uneven sample size comparison. The networks for complex and other diagnoses were very similar, with strong edges being observed in both networks between PSQ1 (*changing sense of self*) and PSQ3 (*others’ views*) as well as PSQ5 (*changing moods*) and PSQ6 (*understandable mood change*). Centrality was highly similar across diagnosis subsamples with PSQ3 (*unstable sense of self*) being the most central item for both those with complex diagnoses (1.55) and those with other diagnoses (1.58). PSQ2 (*others’ views*) was the least central item for both complex and other diagnosis subsamples (−1.45; −1.19 respectively). Predictability was very similar at 0.41 in the complex subsample and 0.37 in the other diagnosis subsample. The NCT showed that there was no significant difference between complex and other diagnosis subsample networks (*p* = 0.39). 

### 3.4. Network Stability

Stability analyses of the global network are reported Figure 3, Figure 4, Figure 5 and Figure 6. Guidance around the CS coefficient for strength centrality recommends that the coefficient should not drop below 0.25 and preferably be above 0.5 to assume that centrality indices are robust [14]. The CS of the global network was 0.75 indicating stable centrality estimates in the full sample. The UK, adult, clinical, complex and common mental health problem networks were stable (0.60, 0.75, 0.67, 0.67 and 0.67 respectively) at the preferable threshold. The Italian (0.28) and community networks (0.48) were stable at the acceptable threshold. The adolescent network was below recommendation (0.21) indicating an unstable network estimation which may be unreliable. 

## 4. Discussion

The present study used a large and sufficiently powered multiple sample and cross-national PSQ dataset to investigate the global network structure of identity disturbance and then assessed whether networks differed across differing subsamples. This investigation of networks across multiple datasets addresses concerns regarding replicability and the overuse of single and non-clinical samples. The PSQ global network suggested that an *unstable sense of self* appeared the most integral aspect of ongoing identity disturbance, followed *changing moods*. These findings are consistent with previous evidence of a lack of continuity in self-perception and affective instability [15] and discontinuities in sense of self and changes in mood [4] as possible maintaining factors of identity disturbance. There was a statistically significant difference in network structures between the UK and Italy and this is in keeping with the theory that identity is an inner construct that is also influenced by society and culture [16]. Network stability analysis showed that the majority of the individually estimated networks were robust. Network comparisons did not significantly differ between adults and adolescents, clinical and community and complex and common presenting problem subsamples. 

This study has shifted the theoretical perspective of the PSQ and that of the structure of identity disturbance. Rather than assuming the PSQ captures a single underlying process of identity disturbance, this study has illustrated how features of identity disturbance may interact. The observed strong edges in the overall network are consistent with theory emphasising that deficits in interpersonal regulation (e.g., self-to other interactions), continuity (e.g., state shifts) and coherence (e.g., mood variability) appear to maintain identity disturbance [4,5]. These networks were found in the context of results demonstrating significant differences between *overall* levels of identity disturbance between national samples, adolescents/adults, clinical/community samples, and complex/common mental health problems. The testing of clinical measures normally is limited to aspects of reliability and validity, and so also adding in network analysis adds methodological and theoretical value to the evaluation of measures. 

### 4.1. Limitations and Future Research

The way complex and common mental health problems were categorised for the purposes of the study could be criticised as being too arbitrary. This research cannot comment on the structure of identity disturbance in non-Westernised countries and so further exploration of the cross-cultural validity and network structure of the PSQ from non-Westernised countries would be useful. Due to the nature of the data available, all adolescent participants were Italian and so the age comparisons were limited. The adolescent subsample was small in comparison to other subsamples and therefore may have lacked power to detect differences between networks. The use of larger, more appropriately matched sample sizes would improve the robustness of future comparisons. The secondary data analysis meant that there was little control over how participants were diagnosed in the original studies and so there was a range of diagnostic uncertainty across the original studies. The inclusion of older adult clinical and community samples would be a useful avenue for future research. In the bootstrapped centrality difference test, the interpretation of each pairwise comparison should be done with caution as the difference test does not control for multiple testing. 

### 4.2. Clinical Implications and Usage

The brevity of the PSQ is appealing in comparison to longer measures of identity disturbance in terms of clinical utility. The PSQ can therefore effectively be integrated into an assessment of the patient and as it is also frequently used in CAT, then this information can then also add into the narrative and diagrammatic reformulation of the patient [5]. As the most central nodes in the overall network (i.e., an unstable sense of self and mood variability) are those most likely to activate other nodes, these may form targets for intervention. Indeed, interventions are being developed that specifically target mood instability and emotion regulation in this manner [17]. The PSQ could potentially track identity integration over time if used as a sessional outcome measure [5] or be able to index when a patient is ‘off-track’ in terms of the stabilisation of state-shifting and self-states. It is also worth noting that interventions need not be developed around centrality alone and also need to consider the individual formulation of the patient. Firstly, nodes may differ in the extent to which they are flexible and susceptible to change through intervention and this will differ from patient to patient. During CAT, there is the adage of ‘push it where it moves’ [5] and therefore targeting pliable modes is a potentially useful clinical approach. So, less central but nodes with getter plasticity and flexibility may therefore have greater potential for change and act as kindling in the network for change in other nodes. It is also possible that the targeting of edges as opposed to individual nodes may be a successful intervention for some patients. In terms of risk management, nodes with the lowest centrality may still be highly important in relation to harm reduction. As state-shifting and identity disturbance is a prominent feature of personality disorder [6], then clinicians are encouraged to use the PSQ is combination with ongoing risk assessments. 

## 5. Conclusions

In summary, the present study has addressed a gap in the evidence base of identity disturbance by providing a network analysis of the PSQ in a large, multi-site and cross-cultural dataset. The findings have clinical implications for the assessment of identity disturbance and have provided avenues for developing potential future interventions. In particular, the use of the PSQ as an outcome measure, the sessional use of the PSQ and also the individualised targeting of edges and nodes according to the case formulation of the patient. Research needs to evaluate in controlled studies the efficacy of such network-driven targeted interventions. In conclusion, the PSQ is emerging as a useful brief measure for assessing and working clinically with identity disturbance. 

## Figures and Tables

**Figure 1 ijerph-19-13793-f001:**
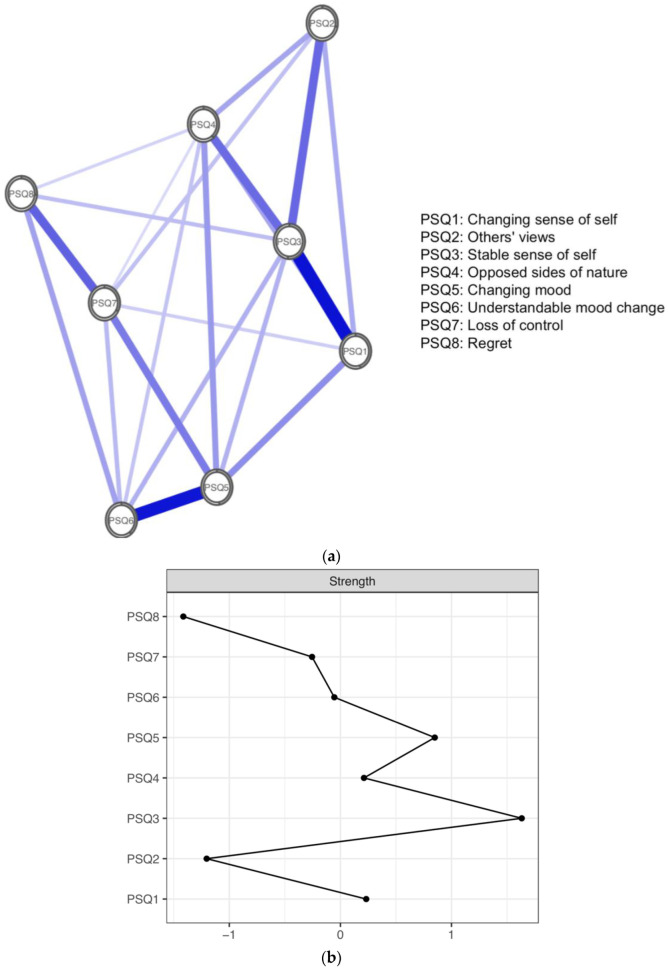
(**a**) Global network of identity disturbance across all datasets. Positive edges are represented by blue lines and shaded areas surrounding nodes represent node predictability. (**b**) Plot of node strength centrality.

**Figure 2 ijerph-19-13793-f002:**
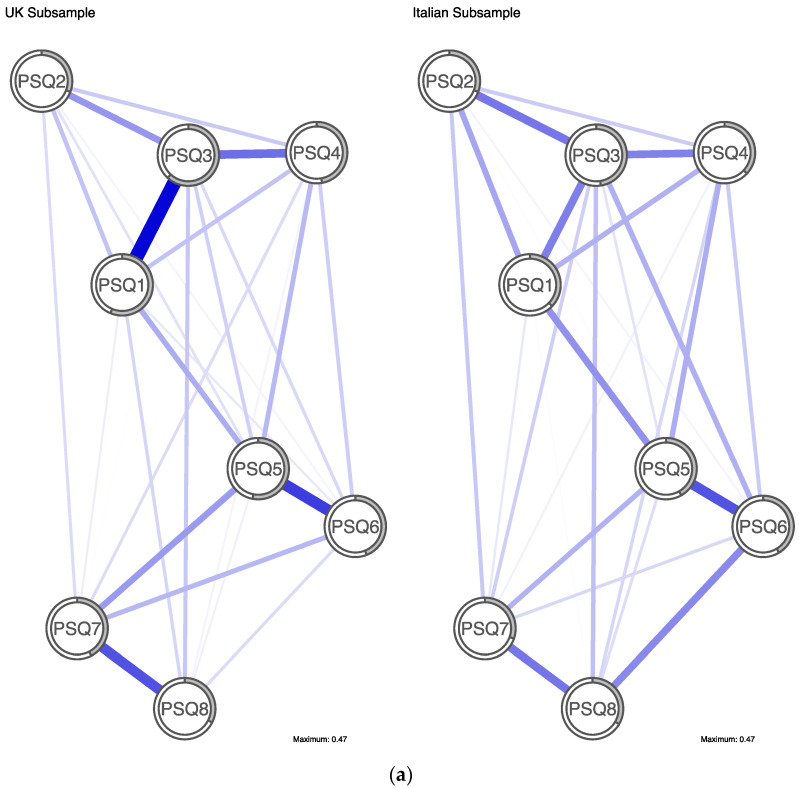
(**a**) Jointly estimated networks of identity disturbance in UK and Italian subsamples. Positive edges are represented by blue lines and shaded areas surrounding nodes represent node predictability. (**b**) Plot of node strength centrality comparison between country subsamples.

**Figure 3 ijerph-19-13793-f003:**
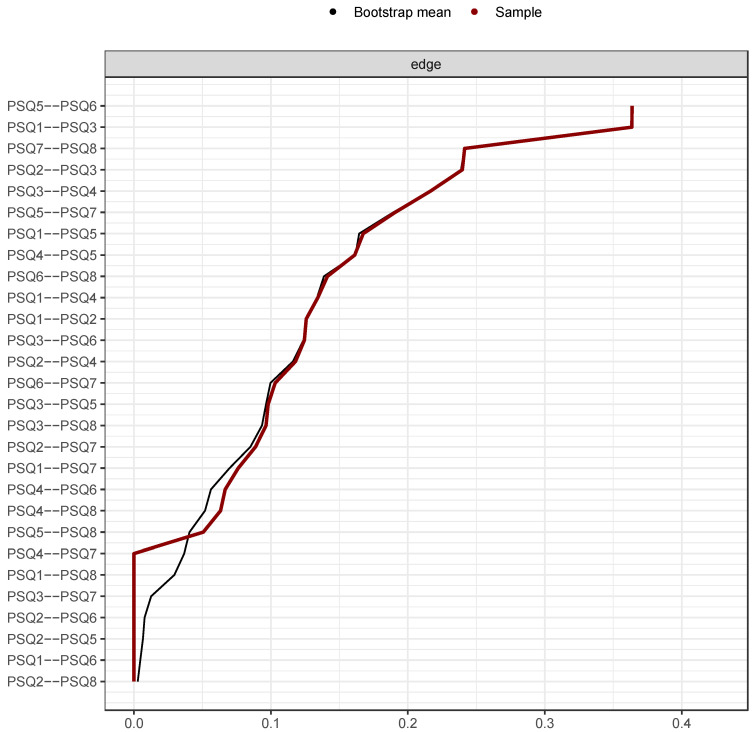
Bootstrapped edge weights for every pairwise node comparison. *Note:* Black line represents bootstrap mean, red line represents point-estimates of each edge weight and the grey shading shows the edge weight 95% confidence intervals.

**Figure 4 ijerph-19-13793-f004:**
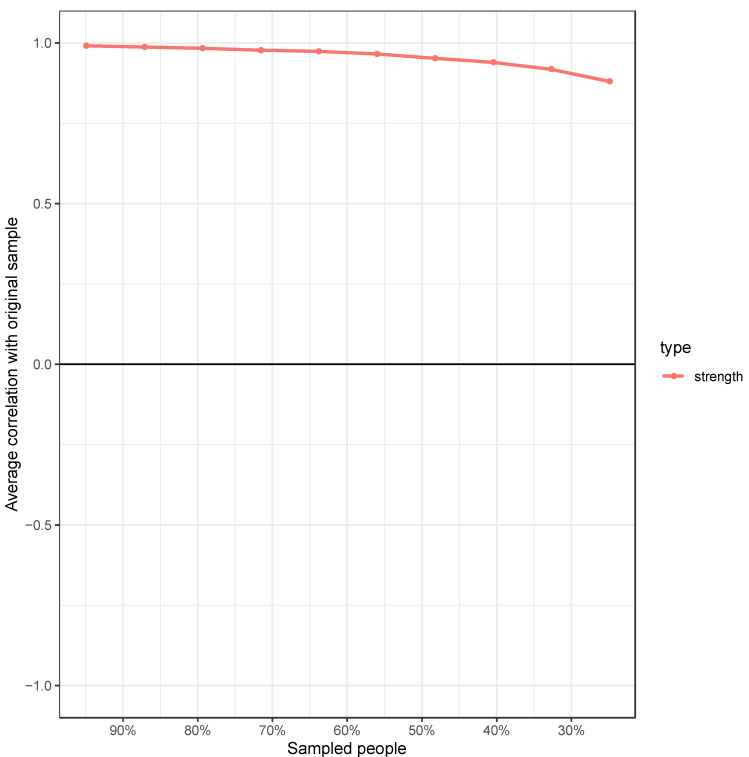
Correlation between the node strength of the PSQ global network and the node strength after randomly dropping a percentage of data from the sample. *Note:* The stability centrality coefficient (CSC) reports the percentage of cases that can be dropped while still retaining 95% certainty of a correlation of 0.7 between the network centrality estimated on the full sample and the networks estimated on subsamples. Guidance suggests a CSC of 0.5 is desirable; the CSC of the global PSQ network = 0.75.

**Figure 5 ijerph-19-13793-f005:**
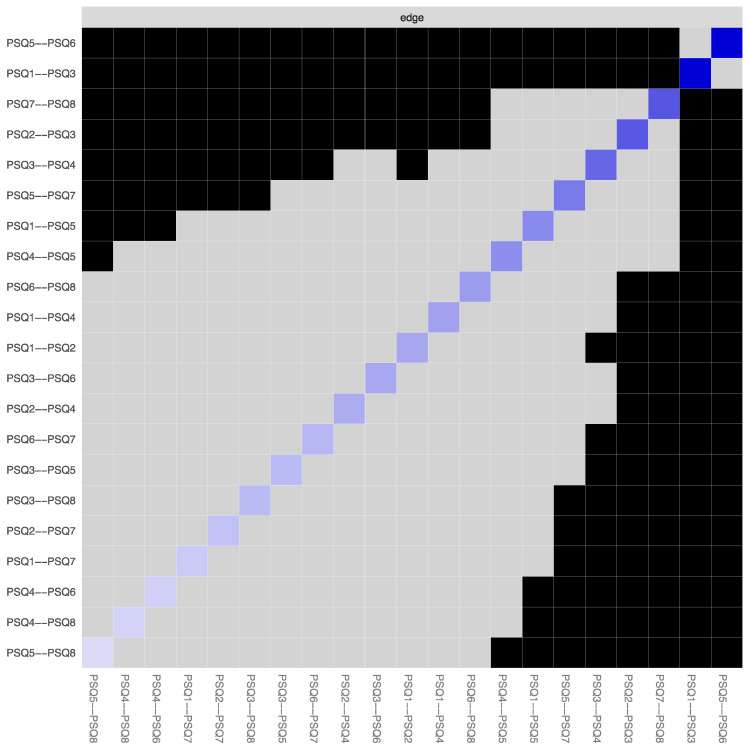
Bootstrapped edge weights difference test. *Note:* This test signifies the bootstrapped significance between pairwise edges in the global PSQ network for every pairwise node comparison (α = 0.05). Significant differences are depicted by black boxes and non-significant differences by grey boxes, while the diagonal-coloured boxes refer to the strength of the edge weight in the network plot (darker blue represents stronger connection).

**Figure 6 ijerph-19-13793-f006:**
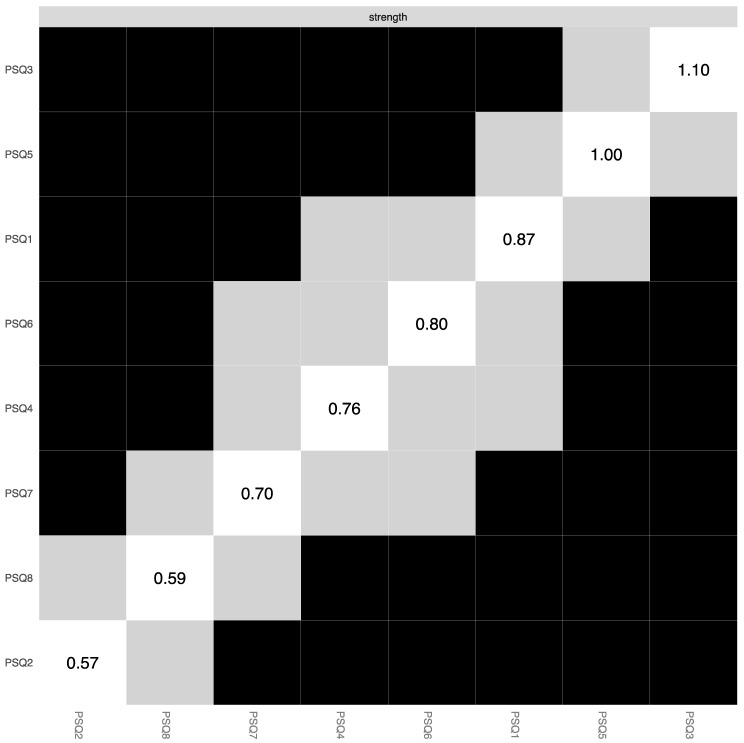
Bootstrapped centrality difference test. *Note:* This tests the bootstrapped significance of centrality estimates for each pairwise node comparison in the global PSQ network (α = 0.05). Significant differences are depicted by black boxes and non-significant differences by grey boxes, while the diagonal boxes refer to the node strength values.

**Table 1 ijerph-19-13793-t001:** Methodological features for each of the PSQ studies making up the sample.

Country (Adult/Adolescent)	Diagnosis or Presenting Problem	PSQ Sample Size	Control Sample Size	Mean Age	Cronbach Alpha Score	Inclusion and Exclusion Criteria	Study Invitation Method	Data Collection Method	Response Rate	Percentage Missing Data
United Kingdom (adult)	Self-harm	96	110	Age recorded as categories. 18–29 years, *n =* 197 30–49 years, *n =* 9	0.83	Clinical group: (1) aged 18 years or older; (2) ≥2 lifetime instances of self-harm; (3) adequate English to complete measures Control group: (1) aged 18 years or older; (2) No lifetime instances of self-harm; (3) Adequate English to complete measures	Clinical group: potential participants referred by clinicians or self-referred in response to adverts. Control group: University study participation system	Clinical group: in person. Control group: online	Unknown, as information not recorded	None
United Kingdom (adult)	Mixed	73	0	Mean 34.84 years (SD = 8.92). Range 18–70)	0.94	Inclusion criteria; seeking psychotherapy in private practice due to mental health problem. Exclusion: none.	Routine practice screen in private practice	Interview	100 %	None
United Kingdom (adult)	Psychosis	182	295	Clinical group mean 33.17 years. Control group mean 25.75 years.	0.87	Clinical group: (1) aged 18 years or older; (2) diagnosis of psychosis or receipt of treatment for psychosis (e.g., antipsychotic medication); (3) Adequate English to complete measures Control group: (1) aged 18 years or older; (2) no diagnosis of psychosis or receipt of treatment for psychosis (e.g., antipsychotic medication); (3) Adequate English to complete measures	Clinical and community samples: adverts placed on social media and mental health websites	Online	Unknown	Clinical group: 25%. Control group: 20%.
United Kingdom (adult)	Mixed	22	0	Mean 37.18 years (SD = 11.19) Range 21–60.	0.87	Inclusion criteria; referred to secondary care mental health care service and receiving cognitive analytic consultancy due to problems with making use of standard service offer.	Routine practice screen in the National Health Service in Secondary Care psychological services	In person interview	100%	None
Italy (adult)	Mixed	237	296	Community mean age 33.36 (SD 13.26) Clinical mean age 32.43 (SD 13.86)	0.85	Inclusion criteria; referred to mental health care services. Exclusion; none.	Clinical: routine practice screen in the Italian public health system. Community; approached local community groups and advertised in local amenities	In person interview for both samples	100% in clinical sample. Unknown in the community sample.	None
Italy (adolescent)	Mixed	152	90	All under the age of 18 years.	0.78	Inclusion criteria; referred to mental health care services. Exclusion; none.	Clinical: routine practice screen in the Italian public health system Community; approached local community groups and advertised in local amenities	Interview for both samples	100% in clinical sample. Unknown in the community sample.	None

**Table 2 ijerph-19-13793-t002:** Demographic characteristics of the full sample.

Variables	*N*	%
Group		
*Clinical*	772	49.7
*Community*	781	50.3
Gender		
*Male*	652	42.0
*Female*	887	57.1
*Unknown*	14	0.9
Age		
*<18*	256	16.5
*18–29*	766	49.3
*30–49*	426	27.4
*50–64*	83	5.3
*>65*	18	1.2
*Unknown*	4	0.3
Nationality		
*UK*	627	40.4
*Italy*	777	50.0
*USA*	58	3.7
*Australia*	8	0.5
*Canada*	10	0.6
*Other–Europe*	30	1.9
*Other–Asia*	21	1.4
*Other–Middle East*	13	0.8
*Other–Central America/Caribbean*	4	0.3
*Other–South America*	3	0.2
*Unknown*	1	0.1
Diagnoses		
*None*	781	50.3
*Depression*	108	7.0
*Self-harm*	39	2.5
*Anxiety disorders, OCD and PTSD*	79	5.1
*Personality disorders (BPD/EUPD)*	30	1.9
*Psychosis*	187	12.0
*Eating and body disorders (AN, BN, BDD, obesity)*	221	14.2
*Developmental disorders (ASD, ADHD)*	58	3.7
*Language disorders*	4	0.3
*Behavioural disorders (including conduct disorder)*	12	0.8
*Long-term physical health condition*	10	0.6
*Substance misuse*	5	0.3
*Other*	13	0.8
*Unknown*	6	0.4

**Table 3 ijerph-19-13793-t003:** Mean item PSQ item score and total PSQ scores with standard deviations (SD) for each subsample.

Item	UK (*n* = 625)	Italian (*n* = 521)	Adult (*n* = 521)	Adolescent (*n* = 254)	Clinical (*n* = 769)	Community (*n* = 780)	Complex Mental Health Problem (*n* = 477)	Common Mental Health Problem (*n* = 1073)
PSQ1	3.12 (1.17)	2.77 (1.17)	2.77 (1.17)	3.31 (1.45)	3.29 (1.23)	2.79 (1.09)	3.25 (1.24)	2.95 (1.15)
PSQ2	2.72 (1.18)	2.59 (1.18)	2.59 (1.18)	3.20 (1.30)	2.98 (1.31)	2.61 (1.10)	2.95 (1.30)	2.72 (1.18)
PSQ3	2.88 (1.16)	2.65 (1.18)	2.65 (1.18)	2.99 (1.28)	3.13 (1.27)	2.58 (1.07)	3.13 (1.28)	2.73 (1.15)
PSQ4	2.90 (1.13)	2.81 (1.18)	2.81 (1.18)	3.16 (1.16)	3.21 (1.21)	2.67 (1.08)	3.18 (1.13)	2.83 (1.14)
PSQ5	3.15 (1.23)	2.85 (1.21)	2.85 (1.21)	3.25 (1.32)	3.38 (1.33)	2.77 (1.11)	3.41 (1.32)	2.92 (1.21)
PSQ6	3.06 (1.28)	2.77 (1.25)	2.77 (1.25)	3.14 (1.37)	3.22 (1.34)	2.72 (1.21)	3.21 (1.34)	2.86 (1.26)
PSQ7	2.76 (1.19)	2.96 (1.11)	2.96 (1.11)	3.35 (1.24)	3.19 (1.25)	2.68 (1.08)	3.17 (1.25)	2.83 (1.16)
PS8Q	3.50 (1.09)	3.19 (1.23)	3.19 (1.23)	3.66 (1.20)	3.67 (1.18)	3.19 (1.15)	3.65 (1.21)	3.33 (1.17)
**Total PSQ**	24.08 (6.92)	22.58 (6.59)	22.58 (6.59)	26.06 (6.30)	26.06 (7.18)	22.01 (5.83)	25.94 (7.24)	23.16 (6.48)

## Data Availability

All the data and the R code for the network analysis are available from the corresponding author on request.

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
