# Peer review of "Modelling Identity Disturbance: A Network Analysis of the Personality Structure Questionnaire (PSQ)"

_ijerph, 2022, doi:10.3390/ijerph192113793_

Round 1

Reviewer 1 Report

The manuscript entitled "Modelling Identity Disturbance: A Network Analysis of the Personality Structure Questionnaire (PSQ)" is an interesting and important work on the associations between identity disturbance in clinical and non-clinical samples of adults from Italy and the UK. Although the study has merit, replication is currently impossible because the Authors did not provide information on how they recruited participants.

1. Please describe step by step in a separate section "Procedure of the study": (1) how the invitation to the study was disseminated; (2) how many people refused (calculate the response rate); (3) what was the form of the study (e.g., online survey, paper-and-pencil, telephone-based, interview, etc.); (4) how many missing data was found and (5) how the Authors deal with this problem; (5) what was exclusion and inclusion criteria? I realize there are 6 samples, so please describe each of them exhaustively to allow replication of this study. 

2. The PSQ is not sufficiently described. Please provide the full response scale with scoring and interpretation for the participant. Please add the information on the reliability of this scale (e.g., Cronbach's alpha) in the original study (with appropriate reference) and in the current study sample.

3. The original scale is developed in English. How were the items translated and adapted to the Italian language? What were the parametric properties of the PSQ in the Italian sample? Was one-factor structure replicated?

4. Please add the subsection describing all demographic variables and all options of response for each variable.

5. Please report the results of the Student's t-test according to scientific standards, providing information for the t-test, df, and p-value. Please see the APA style guideline to learn how these statistics should be reported in the text. For example, instead of "(t, 773) =", this statistic should be reported as  "t(773) =". The effect size (e.g., Cohen's d) is missing at all and should be added for each comparison.

6. Conclusion should be extended on some practical implications. The Authors stated: "The findings have clinical implications for the assessment of identity disturbance and have provided avenues for developing potential future interventions." Please add examples of how the results of this study can be used in clinical practice (giving various types of intervention).

Author Response

  1. Please describe step by step in a separate section "Procedure of the study": (1) how the invitation to the study was disseminated; (2) how many people refused (calculate the response rate); (3) what was the form of the study (e.g., online survey, paper-and-pencil, telephone-based, interview, etc.); (4) how many missing data was found and (5) how the Authors deal with this problem; (5) what was exclusion and inclusion criteria? I realize there are 6 samples, so please describe each of them exhaustively to allow replication of this study. 

We have included a new table 1 for the paper that describes these features for each study and so this also enables a cross-study comparison.   

  1. The PSQ is not sufficiently described. Please provide the full response scale with scoring and interpretation for the participant. Please add the information on the reliability of this scale (e.g., Cronbach's alpha) in the original study (with appropriate reference) and in the current study sample.

We have added a section in which the PSQ is more fully described.  The alpha scores have been added to the new Table 1. 

  1. The original scale is developed in English. How were the items translated and adapted to the Italian language? What were the parametric properties of the PSQ in the Italian sample? Was one-factor structure replicated?

The translation methods have been added to the method.  Table 1 contains the alpha scores for the Italian samples.  The single factor nature of the PSQ is covered in the introduction.      

  1. Please add the subsection describing all demographic variables and all options of response for each variable.

This section has been added to the method.

  1. Please report the results of the Student's t-test according to scientific standards, providing information for the t-test, df,and p-value. Please see the APA style guideline to learn how these statistics should be reported in the text. For example, instead of "(t, 773) =", this statistic should be reported as  "t(773) =". The effect size (e.g., Cohen's d) is missing at all and should be added for each comparison.

The document has been checked for APA formatted reporting. 

  1. Conclusion should be extended on some practical implications. The Authors stated: "The findings have clinical implications for the assessment of identity disturbance and have provided avenues for developing potential future interventions." Please add examples of how the results of this study can be used in clinical practice (giving various types of intervention).

A new section on clinical implications has been added as this is in keeping with the other reviewer’s wishes.  A separate conclusion section has also been added.   

Reviewer 2 Report

This paper reports on a network analysis of the Personality Structure Questionnaire. The findings are well presented, novel and have some possible clinical applications.

Author Response

Thank you for the very positive feedback  

Round 2

Reviewer 1 Report

The Authors improved the manuscript, but remain problems, which prevent the manuscript from publication:

1. It is unclear, how the differences between subsamples were assessed. The "Data analysis" section should clearly inform, what statistical test will be used (e.g., ANOVA or independent samples t-test), how effect size will be measured, and how it should be interpreted. 

2. The statistics in the "Demographics and sample description" section are unbelievable. First of all, it is unclear why two numbers (1 and 144) refer to one df in the Student's t-test. Secondly, the degree of freedom cannot be "(1,144)" since the UK sample included 625 participants, and the Italian sample consisted of 521 individuals (see page 5, line 32). This is only an example because all statistics on page 3 lines 117-122 are completely not probable. 

3. The heading of Table 3 should include the number of people for each sample.

Author Response

  1. It is unclear, how the differences between subsamples were assessed. The "Data analysis" section should clearly inform, what statistical test will be used (e.g., ANOVA or independent samples t-test), how effect size will be measured, and how it should be interpreted. 

Response: A sentence describing the statistical test and effect size measures used to compare mean PSQ scores between subsamples has been added to the data analysis section. 

  1. The statistics in the "Demographics and sample description" section are unbelievable. First of all, it is unclear why two numbers (1 and 144) refer to one df in the Student's t-test. Secondly, the degree of freedom cannot be "(1,144)" since the UK sample included 625 participants, and the Italian sample consisted of 521 individuals (see page 5, line 32). This is only an example because all statistics on page 3 lines 117-122 are completely not probable. 

Response: The df's do not refer to two numbers (e.g. 1 and 144) but to 1,144 - which correspond to the sample sizes of the respective subsamples (n-2). These have been edited to make this clear. 

  1. The heading of Table 3 should include the number of people for each sample.

Response: The subsample n's have been added to the heading of Table 3.